# Pressure-stabilized divalent ozonide CaO$_3$ and its impact on Earth's oxygen cycles

Yanchao Wang [1], Meiling Xu [2], Liuxiang Yang [3], Bingmin Yan[3], Qin Qin[3], Xuecheng Shao[1], Yunwei Zhang [1], Dajian Huang[3], Xiaohuan Lin[3], Jian Lv[1], Dongzhou Zhang [5], Huiyang Gou [3✉], Ho-kwang Mao[3,4], Changfeng Chen [6✉] & Yanming Ma [1,7✉]

High pressure can drastically alter chemical bonding and produce exotic compounds that defy conventional wisdom. Especially significant are compounds pertaining to oxygen cycles inside Earth, which hold key to understanding major geological events that impact the environment essential to life on Earth. Here we report the discovery of pressure-stabilized divalent ozonide CaO$_3$ crystal that exhibits intriguing bonding and oxidation states with profound geological implications. Our computational study identifies a crystalline phase of CaO$_3$ by reaction of CaO and O$_2$ at high pressure and high temperature conditions; ensuing experiments synthesize this rare compound under compression in a diamond anvil cell with laser heating. High-pressure x-ray diffraction data show that CaO$_3$ crystal forms at 35 GPa and persists down to 20 GPa on decompression. Analysis of charge states reveals a formal oxidation state of −2 for ozone anions in CaO$_3$. These findings unravel the ozonide chemistry at high pressure and offer insights for elucidating prominent seismic anomalies and oxygen cycles in Earth's interior. We further predict multiple reactions producing CaO$_3$ by geologically abundant mineral precursors at various depths in Earth's mantle.

[1] State Key Lab of Superhard Materials & International Center for Computational Method and Software, College of Physics, Jilin University, Changchun 130012, China. [2] School of Physics and Electronic Engineering, Jiangsu Normal University, Xuzhou 221116, China. [3] Center for High Pressure Science and Technology Advanced Research, Beijing 100094, China. [4] Geophysical Laboratory, Carnegie Institution of Washington, Washington, DC 20015, USA. [5] Hawai'i Institute of Geophysics and Planetology, School of Ocean and Earth Science and Technology, University of Hawai'i at Manoa, Honolulu, HI 96822, USA. [6] Department of Physics and Astronomy, University of Nevada, Las Vegas, NV 89154, USA. [7] International Center of Future Science, Jilin University, Changchun 130012, China. ✉email: huiyang.gou@hpstar.ac.cn; chen@physics.unlv.edu; mym@jlu.edu.cn

Pressure and temperature are key thermodynamic variables that prominently influence material structure and properties. Diverse high-pressure and high-temperature (HPHT) conditions simulated in computation and generated in laboratory-based experimental devices offer exciting opportunities for new material discovery and exploration of otherwise inaccessible deep-Earth environments. Recent years have seen the advent and rapid advance of computational structure search and characterization of pressure-stabilized compounds with unusual stoichiometries, such as Na-Cl[1], Xe-Fe[2], Xe-O[3], and La-H[4] series that do not exist at ambient conditions, and several of these compounds have already been experimentally synthesized[5,6]. Also notable are recent experimental and theoretical studies that have led to the discovery of unconventional iron oxides with unusual oxidation states[7] in FeO_2 (ref. [8]), $Fe_2O_3$ (ref. [9]), and $Fe_5O_6$ (ref. [10]), opening avenues for making and exploring iron oxides with peculiar properties like unusual chemical valence and bonding interactions. Such results also restore the redox equilibria inside Earth and place oxygen reservoirs at greater depths than previously thought. These findings offer insights for elucidating large-scale geological activities that may have influenced events related to the origin of life on Earth.

Calcium and oxygen are two of the most abundant elements widely distributed in Earth's mantle[11]. In accordance with their respective valence electron counts, calcium and oxygen are expected to preferably form CaO[12], which is present throughout the mantle. The oxidation state of O in CaO is $O^{2-}$ at ambient conditions. Meanwhile, calcium peroxide[13,14] also can stabilize at ambient or high pressures[15]. In general, oxygen species with oxidation states higher than $-2$ can be synthesized in superoxide ($O_2^-$), peroxide ($O_2^{2-}$), and ozonide ($O_3-$) compounds and play prominent roles in oxidation chemistry[16]. Among these compounds, ionic ozonides are regarded as a species with unusual reaction processes and properties, and are scarce due to their high reactivity, thermodynamic instability, and extreme sensitivity to moisture in ambient environments[17].

In this work, we report on the discovery from a combined theoretical and experimental study a high-pressure phase of $CaO_3$ containing unusual divalent ozone anions that shed light on ozonide chemistry at extreme conditions, and the results offer insights for understanding deep-Earth chemical reactions that are relevant to oxygen cycles inside our planet.

## Results and discussion

**Stable Ca–O compounds at high pressure.** For insights to help find new calcium oxide compounds, we have employed unbiased crystal structure search techniques as implemented in CALYPSO code[18,19], which has been successful in resolving crystal structures of a large number and variety of materials at high pressure[20]. Here, we explore calcium oxides in the oxygen-rich regime, seeking compounds that do not exist under ambient conditions. Studies of mantle rocks have shown that oxygen fugacity of the upper mantle is relatively high[21], thus connecting the present work to prominent geological topics concerning oxidation states of minerals and oxygen storage and cycles inside Earth. We have performed structure searches on $Ca_mO_n$ ($m = 1, 2$ and $n = 2, 3, 4$) with maximum simulation cells up to four formula units (f.u.) at each composition, and this procedure identifies two stable Ca–O compounds, a $CaO_2$ phase at 30 GPa and an unusual stoichiometric $CaO_3$ phase at 50 GPa. This result distinguishes $CaO_2$ and $CaO_3$ as two viable oxygen-rich calcium oxides. Furthermore, the $CaO_4$-containing superoxide group ($O_2^-$) is found to stay above but close to the convex hull, making it energetically more favorable with respect to the dissociation route into $CaO + O_2$ above 34 GPa and thus may be experimentally synthesized at high temperature [see Supplementary Note and Supplementary Figs. 1–3 for details on the structure search results].

We characterize the newly identified calcium ozonide by examining its synthesis routes and structural, bonding, and electronic properties. The $CaO_3$ phase crystalizes[22] in a tetragonal $BaS_3$-type structure[23] (space group $P$-$42_1m$, 2 f.u. per cell) in a wide range of pressures and exhibits a distinct configuration containing isolated V-shaped $O_3$ units and edge-sharing $CaO_8$ cuboid (Fig. 1a). We compare to some well-established compounds on key structural and bonding characters of the crystalline $CaO_3$ at 30 GPa, which is inside its stability field. The O–O bond length and O–O–O bond angle of the ozonide anion in $CaO_3$ are 1.44 Å and 114.57°, respectively, larger than the corresponding values in $KO_3$ (1.34 Å, 109.33°)[16], indicating a weaker O–O bonding and more ionic nature due to additional electrons in the antibonding molecular orbitals (MO) of ozonide anion. The same pattern of relative bond lengths and angles for $CaO_3$ and $KO_3$ persists in a wide pressure range of 20–50 GPa (see Supplementary Fig. 4). Moreover, the shortest Ca–O distance in $CaO_3$ (2.31 Å) is comparable to that in the prototype ionic compound CaO (2.27 Å)[12], suggesting an ionic bonding between Ca and $O_3$ in $CaO_3$.

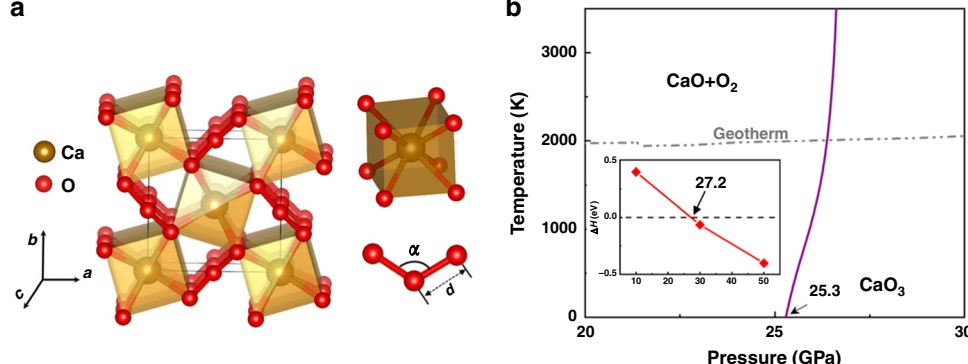

**Fig. 1 Crystal structure and phase stability of $CaO_3$. a** Crystal structure of the newly identified tetragonal phase of $CaO_3$. **b** Thermodynamic phase diagram of $CaO_3$ determined using first-principles density functional theory, including the proton zero-point motion at the harmonic level, highlighting the phase boundary for decomposition into CaO and $O_2$ in reference to the geotherm line. The purple line is the phase boundary for $CaO_3$ decomposition into CaO and $O_2$. The gray dotted line presents the geotherm of the Earth from ref. [48]. The inset shows calculated formation enthalpy of $CaO_3$ with respect to decomposition into CaO and $O_2$ as a function of pressure at zero temperature.

Formation enthalpy calculations reveal that $CaO_3$ is energetically favorable relative to decomposition into $CaO$[12] and solid $O_2$ (ref. [24]) above 27.2 GPa via the reaction:

$$CaO + O_2 = CaO_3 \qquad (1)$$

To account for thermal effects, we further examine vibrational contributions and entropic effects for the relevant phases[25], and construct the finite-temperature phase diagram of $CaO_3$. Calculated zero-point energy values at 30 GPa for $CaO$, solid $O_2$, and $CaO_3$ are 0.13, 0.22, and 0.32 eV/f.u., respectively, resulting in a minimal difference between the reactants and products for the above reaction of only $-0.03$ eV/f.u., which has only a minor impact on the threshold pressure for $CaO_3$ decomposing into $CaO$ and $O_2$, reducing it from 27.2 to 25.3 GPa (Fig. 1b). The threshold pressure for the stability of $CaO_3$ increases with rising temperature, going from 25.3 GPa at 0 K to 26.5 GPa at 2000 K. We further checked phonon dispersions of $CaO_3$ at 20 and 50 GPa, and the results (see Supplementary Fig. 5) show no imaginary frequencies, indicating that $CaO_3$ is dynamically stable in this wide pressure range, making it a metastable phase at lower pressures like diamond versus graphite.

**Experimental synthesis of Ca–O compounds**. Our systematic assessment of energetic, dynamic, and thermodynamic stability of $CaO_3$ under pressure suggests its likely synthesis through the reaction indicated by Eq. (1). We have performed HPHT experiments employing a laser-heated diamond anvil cell (DAC). $CaO$ or Ca powder and liquefied $O_2$ were loaded into an Re or Fe gasket hole, then compressed gradually to 35–40 GPa, and heated up to a temperature of approximately 3100 K with an off-line laser heating technique (see "Methods"). The Raman spectra were collected during the laser heating as shown in Fig. 2a, where new features are presented in the Raman spectra of the laser heated $CaO$ or Ca and $O_2$, unlike those of the pure solid oxygen appeared within the sample chamber[22]. Distinct new Raman modes are observed at about 767 and 1140 cm$^{-1}$, suggesting the formation of new phases. The low-frequency modes appeared at about 200 cm$^{-1}$ also indicate the presence of potential new bonding structure distinct from the starting $CaO$ and $O_2$. The calculated Raman spectrum of $CaO_3$ at 38 GPa is found to be close to those modes observed in the experiment. In particular, the frequency mode near 767 cm$^{-1}$ observed in experiment can be assigned to the vibration mode of $O_3^{2-}$, whereas the high-frequency mode of 1140 cm$^{-1}$ is attributed to the vibration mode of intramolecular O–O in $CaO_4$, whose bond length of 1.31 Å is within the length range (1.3–1.4 Å) of the superoxide $O_2^-$.

The powder X-ray diffraction (PXRD) patterns around the heating spot were collected are shown in Fig. 2b. We observe two distinct Bragg peaks at 10.6° and 11.4° and several small peaks from the raw 2D diffraction images and integrated PXRD patterns that do not correspond to $CaO$, $CaO_2$, or any known calcium oxides. Meanwhile, the measured XRD pattern can be indexed by the predicted tetragonal $BaS_3$-type structure of $CaO_3$, together with $CaO_4$, unreacted $CaO$, and oxygen, due to the mixed feature of obtained phases. The observed peaks at 10.6° and 11.4° in the XRD pattern correspond to the (200) and (111) crystal planes of tetragonal $CaO_3$. The obtained lattice parameters of the synthesized $CaO_3$, $a = 4.67$ Å, and $c = 2.92$ Å, are close to theoretical data of $a = 4.87$ Å and $c = 2.98$ Å. A decompression run was performed to assess volume change versus pressure, and the resulting pressure–volume data (Supplementary Fig. 6) are fitted by equation of state with $B_0 = 103(9)$ GPa, $B_0' = 3.9$ for $CaO$ and $B_0 = 114(11)$ GPa, $B_0' = 2.7$ for $CaO_3$, in good agreement with theoretical $B_0$ values, 113.6 GPa, $B_0' = 4.0$ for $CaO$, and 99.8 GPa, $B_0' = 4.0$ for $CaO_3$, which are obtained by fitting calculated total energies versus volume to the Birch–Murnaghan equation[26]. The $CaO_3$ signals persist to at least 20.0 GPa as shown in Supplementary Fig. 7a.

**Electronic properties**. To decipher the nature of bonding and charge states in $CaO_3$, we have examined electron localization function[27] in this compound. The results (Fig. 3a, upper panel) show clear covalent O–O bonding evidenced by the strong charge localization between the nearest-neighbor O atoms in the $O_3$ units. Meanwhile, a less localized charge distribution is seen on the asymmetric Ca–O bonds (Fig. 3a, lower panel), indicating a significant degree of ionicity between the $O_3$ anions and Ca cations. From a Bader charge analysis[28], charge values on Ca and O are calculated at 30 GPa, and the results are listed in Table 1. There is a charge transfer of 1.51$e$ from Ca to $O_3$ unit in $CaO_3$,

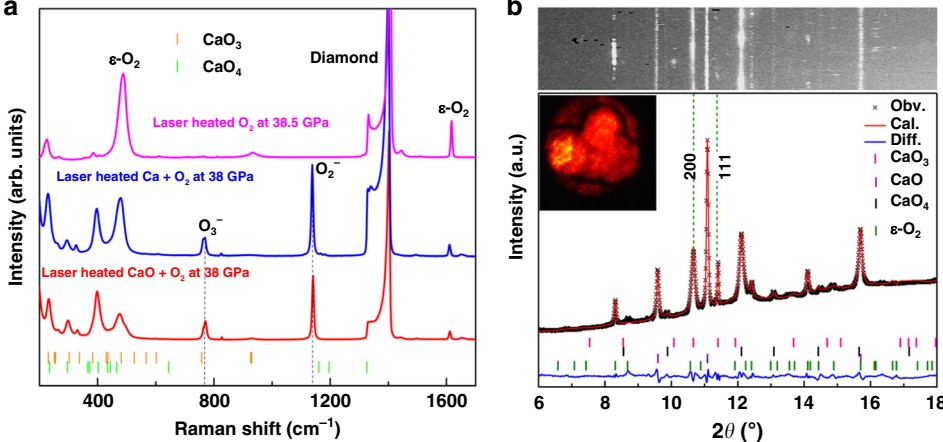

**Fig. 2 Raman spectra and X-ray diffraction pattern of CaO$_x$. a** Raman spectra of Ca–O compounds at high pressure. The calculated frequencies of Raman active vibrational modes are indicated by vertical bars. **b** Measured powder X-ray diffraction pattern of Ca–O compounds at 35 GPa with the Rietveld method (XRD 2D image is shown at the top; inset shows a microphotographic image in the gasket hole of about 100 μm through diamond culets). Vertical ticks correspond to the Bragg peaks of $CaO_3$ (pink), $CaO_4$ (orange), $CaO$ (purple), and solid $O_2$ (wine). The X-ray wavelength is 0.4337 Å. The obtained lattice pentameters are $a = 4.11$ Å, $c = 5.04$ Å for tetragonal $CaO_4$, $a = 4.49$ Å for cubic $CaO$, and $a = 7.13$ Å, $b = 4.57$ Å, $c = 3.74$ Å, $\beta = 110.2°$ for monoclinic oxygen.

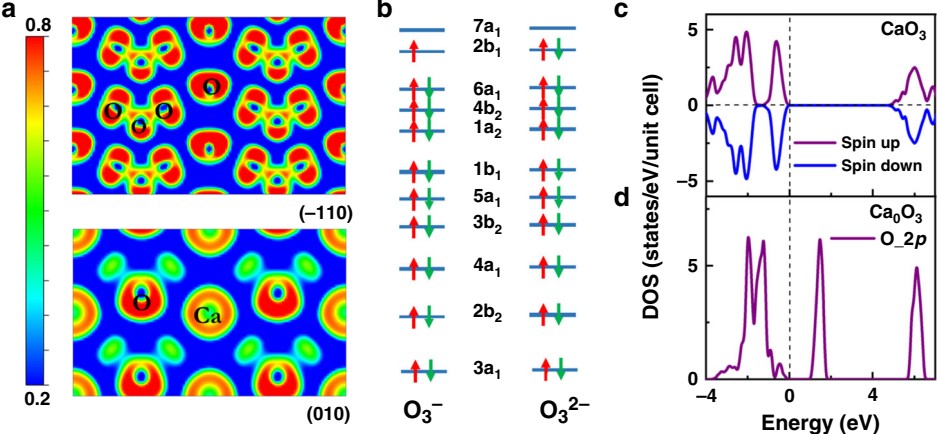

**Fig. 3 Charge and magnetic order in CaO$_3$. a** Calculated 2D electron localization function of CaO$_3$ plotted in the (−110) and (010) planes at 30 GPa. **b** The molecular orbitals-scheme for O$_3^{1-}$ and O$_3^{2-}$, following the sequence in refs. [29,49]. The red and green arrows represent spin-up and spin-down electrons, respectively. **c** Density of states (DOS) of CaO$_3$ at 30 GPa and **d** The O 2p states for hypothetical Ca$_0$O$_3$, which exhibits partially unoccupied bonding states between 1 and 2 eV that become filled by electrons transferred from Ca in CaO$_3$. The vertical dashed line indicates the position of the Fermi energy.

**Table 1 Bader charge analysis of CaO$_3$, KO$_3$, CaO, and CaO$_2$ at 30 GPa.**

| Compounds | Ca/K (e) | O$_{bridge}$(e) | O$_{terminal}$(e) | O (e) |
|---|---|---|---|---|
| CaO$_3$ | 1.51 | −0.21 | −0.65 | |
| KO$_3$ | 0.86 | +0.04 | −0.45 | |
| CaO | 1.43 | | | −1.43 |
| CaO$_2$ | 1.48 | | | −0.74 |

comparable to those in CaO (1.43$e$) and CaO$_2$ (1.48$e$), but much greater than that in KO$_3$ (0.86$e$). This result highlights a crucial distinction of the O$_3$ unit in CaO$_3$ compared to the [O$_3$]$^{-1}$ anion in KO$_3$. It is also seen that the two terminal O atoms carry more negative partial charges (0.65$e$ per O) than the bridge O atom (0.21$e$) within each O$_3$ anion in CaO$_3$, because the central O interacts much less with Ca cations than with the terminal O anions. To the best of our knowledge, the presently identified [O$_3$]$^{-2}$ ionic charge state has never been seen in other ozonides[17].

We examine MO schemes[29,30] to elucidate electronic configurations in O$_3^-$ and O$_3^{-2}$. Results (Fig. 3b) show that electrons in the antibonding 2b$_1$ orbital dictate properties of O$_3$ anions. Alkali-metal ozonides[29] containing [O$_3$]$^-$ belong to a small group of chemical species hosting unpaired $p$-electrons that produce a paramagnetic state. In stark contrast, divalent ozonide anion [O$_3$]$^{-2}$ has a closed-shell configuration (Fig. 3b) with each O$_3$ unit containing 20 electrons in 10 orbitals with no unpaired $p$-electron, leading to non-magnetic characteristics (Fig. 3c). To illustrate this point, we have constructed a model system of hypothetical Ca$_0$O$_3$ where all Ca atoms were removed from the BaS$_3$-type structure, and this model system exhibits partially unoccupied bonding states of the O 2p orbital (Fig. 3d), which become fully occupied once Ca was incorporated into the crystal lattice due to charge transfer from Ca to O, leading to the non-magnetic insulating state in CaO$_3$.

**Implications for geoscience**. The reactivity of CaO and O$_2$ is strongly driven by the denser structural packing of CaO$_3$. Our calculations show that the reaction indicated in Eq. (1) at 30 GPa supports volumes of 23.48 Å$^3$ for CaO in $Fm$-$3m$ structure, 15.86 Å$^3$ for O$_2$ in $Cmcm$ structure, and 36.20 Å$^3$ for CaO$_3$, with a large volume shrinkage of $\Delta V/V = -7.96\%$. Consequently, the

PV term in Gibbs free energy strongly favors the formation of CaO$_3$ at high pressures. For comparison, we also have explored the possibility of forming MgO$_3$ at pressures up to 50 GPa, but the associated positive formation enthalpy (1.45 eV/f.u.) and smaller volume shrinkage of −3.76% render MgO$_3$ unstable against decomposition into MgO and solid O$_2$ in $C2/m$ symmetry, in agreement with previous reports[31].

The newly discovered calcium ozonide is expected to have major implications for geoscience. In this context, we have examined additional viable routes producing CaO$_3$ involving several minerals abundant in Earth's mantle as reactants:

$$Ca(OH)_2 + O_2 = H_2O + CaO_3, \qquad (2)$$

$$Ca(OH)_2 + Al_2O_3 + O_2 = 2AlOOH + CaO_3, \qquad (3)$$

$$4FeO_2 + CaO + H_2O = 4FeOOH + CaO_3 \qquad (4)$$

with the structures of pertinent materials employed in calculating the reaction enthalpies are presented in Supplementary Table 1. Similar to the reaction shown in Eq. (1), the reactions in Eqs. (2) and (3) occur in oxygen-saturated environments and the reactants and products attain equilibrium at pressures of 20 GPa (Fig. 4a) and 40 GPa (Fig. 4b), respectively, corresponding to conditions near the top of the lower mantle, where previous studies reveal that oxygen fugacity is likely inhomogeneous with some regions containing relatively high oxygen content[21], thus conducive to these reactions in forming CaO$_3$.

Our calculations show that the reactions described in Eqs. (1) and (2) produce CaO$_3$ at ~20 GPa, which corresponds to pressures at the boundary of Earth's upper and lower mantle. Previous studies revealed that several minerals such as CaCO$_3$ (ref. [32]), MgCO$_3$ (ref. [33]), and CO$_2$ (ref. [34]) can dissociate and produce oxygen in this pressure range, offering an abundant source of O$_2$ for these proposed reactions inside Earth's mantle. The resulting compound CaO$_3$, which was not previously considered, provides an alternative mechanism to explain seismic anomalies near 660 km depth in Earth's mantle where pressure is ~20 GPa[35,36].

The reaction route indicated in Eq. (4) describes the formation of CaO$_3$ and FeOOH by FeO$_2$ and CaO under H$_2$O-saturated conditions and the equilibrium pressure of this reaction is about 90 GPa (Fig. 4c), corresponding to deep lower mantle conditions.

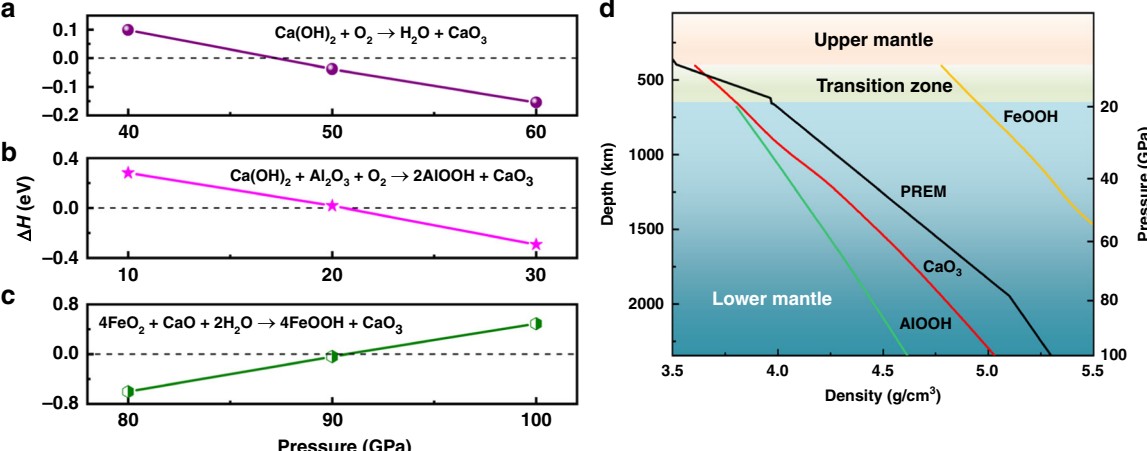

**Fig. 4 Phase equilibria and the density of minerals related to production of CaO₃. a–c** Relative enthalpy of proposed reactions forming CaO₃ at high pressure. **d** Comparison of the density of FeOOH, AlOOH, and CaO₃ with that of Earth's mantle according to the Preliminary Reference Earth Model (PREM)[37].

Recently, a pyrite $FeO_2$ phase stabilized at high pressure (76 GPa) and temperature (1800 K) was proposed[8] to exist in Earth's lower mantle below 1800 km. Our calculations show that once CaO and $H_2O$ are thrusted to deeper than 1800 km, they can react with $FeO_2$ and produce $CaO_3$ + FeOOH. Due to the higher density of FeOOH compared to that of the mantle[37] (Fig. 4d), FeOOH would sink towards the core[38], while the lighter $CaO_3$ would ascend by mantle dynamic processes. Once reaching the transition zone at depths of less than 500 km, $CaO_3$ would decompose to provide a sporadic source of extra $O_2$ that would work its way up toward the surface of Earth to complete the oxygen cycle.

Oxygen fugacity and oxidation states of minerals in geological environments play pivotal roles in deciding planetary chemical and physical dynamics, and such key information can be determined through mineral equilibria[39]. Quantification of oxygen fugacity depends sensitively on the content and stability of mineral assemblages at the pressures and temperatures in Earth's interior. Our discovery of divalent ozonide $CaO_3$ introduces a new ingredient to buffer oxygen fugacity and influence redox equilibria of Earth's mantle, providing crucial insights into the redox state of the largely inaccessible deeper mantle. Furthermore, our results highlight CaO as a reducing agent to react with free oxygen at high pressures, suggesting a natural reservoir for $O_2$ storage in Earth's mantle and providing a possible resolution to the missing $O_2$ paradox before the Great Oxidation Event[40]. The present findings also raise exciting prospects of synthesizing $CaO_3$ via additional avenues, such as those listed in Eqs. (2)–(4), in the laboratory setting for a more in-depth understanding of these reactions and their roles in influencing important geological events. The discovery of crystalline divalent calcium ozonide is expected to stimulate further experimental and theoretical exploration for further insights into this compound and the associated intriguing bonding characters that hold great promise for probing exotic properties that have great fundamental significance and implications for practical processes in chemistry and geoscience.

We have conducted a joint computational and experimental exploration of calcium oxides at high pressure, aiming to probe unusual stoichiometry, structural form, and oxidation states. Our study leads to a discovery of $CaO_3$, expanding both the calcium oxide family and ionic ozonide family of compounds. This rare crystalline ozonide is computationally predicted and then experimentally synthesized via reaction of solid CaO and $O_2$ at

HPHT conditions in a DAC assisted by laser heating. Remarkably, a charge analysis indicates that the $O_3$ unit in $CaO_3$ carries a formal oxidation state of −2. These findings enrich fundamental understanding of bonding interactions between calcium and oxygen, highlighting novel ozonide chemistry at high pressure, and the reported results have major implications for elucidating prominent seismic anomalies and oxygen cycle processes in Earth's mantle.

## Methods

**Experimental procedures.** High-purity CaO (Alfa, 99.95%) powder or Ca piece (Alfa, 99%) were compressed into thin plates of 50 μm × 50 μm × 15 μm dimensions and loaded in a DAC with a culet of 300 μm. The sample chamber has a 100 μm diameter hole drilled in a pre-indented rhenium or steel gasket (38 μm thickness). The DAC was placed in a sealed container immersed in liquid nitrogen. $O_2$ gas (99.999%) was piped into the container. Liquefied $O_2$ infused into the sample chamber as the pressure medium and precursor. The samples were pressurized to 35–40 GPa and heated up to ~3100 K by an offline double-sided laser-heating (wavelength 1064 nm) system at HPSTAR and HPSynC of the Advanced Photon Source (APS), Argonne National Laboratory. Temperature was obtained from fitting the thermal radiation spectra to the Planck radiation function right after the reported chemical reaction has occurred in the DAC sample chamber. Laser spots at HPSTAR and HPSynC are approximately 20 μm in diameter. Pressure was calibrated by the fluorescence of ruby balls placed inside the sample chamber[41]. Optical absorption was monitored during and after the compression process. Synchrotron XRD data were also collected at 35–40 GPa and during the ensuing decompression process at the 13-BMC (λ = 0.4337 Å), GeoSoilEnviroCARS, Argonne National Laboratory and BL15U1 at Shanghai Synchrotron Radiation Facility (λ = 0.6199 Å). The X-ray probing beam size was about 15 μm at the bending beamlines, and 5 μm at the undulator beamlines.

**Ab initio calculations.** Our structure prediction is performed using CALYPSO (Crystal structure AnaLYsis by Particle Swarm Optimization) methodology[18,42] as implemented in its same-name CALYPSO code[19] (CALYPSO code is free for academic use, by registering at http://www.calypso.cn.), which is based on a global minimization of free energy surfaces in conjunction with ab initio total-energy calculations. Structural optimization, electronic structure, and phonon calculations were performed in the framework of density functional theory within the generalized gradient approximation[43] as implemented in the VASP code[44]. The electron–ion interaction was described by the projector augmented-wave potentials[45], with $3s^23p^64s^2$ and $2s^22p^4$ configurations treated as the valence electrons of Ca and O, respectively. The dynamic stability of the predicted new phases was verified by phonon calculations using the direct supercell method as implemented in the PHONOPY code[46]. Crystal structures were visualized with VESTA[47].

## Data availability

The authors declare that the main data supporting the findings of this study are contained within the paper and its associated Supplementary Information. All other relevant data are available from the corresponding authors upon reasonable request.

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

## Acknowledgements

Y.W. and Y.M. acknowledge funding support from the National Key Research and Development Program of China under Grant No. 2016YFB0201201 and No. 2017YFB0701503; the National Natural Science Foundation of China (NSFC) under Grants No. 11774127, 11822404, 11904142, and 11534003; supported by Program for JLU Science and Technology Innovative Research Team (JLUSTIRT); and the Science Challenge Project, No. TZ2016001. H.G. acknowledges support from the NSFC under Grants No. U1530402. Part of the calculation was performed in the high-performance computing center of Jilin University and at Tianhe2-JK in the Beijing Computational Science Research Center.

## Author contributions

Y.W., H.G., and Y.M. conceived and designed the project and directed the calculations and experiments. Y.W., M.X., X.S., Y.Z., J.L., and Y.M. performed computer simulations. L.Y., B.Y., Q.Q., X.L., D.Z., H.G., and H.K.M. performed experimental measurements. Y.W., H.G., C.F.C., Y.M., and H.K.M. analyzed the results and wrote the manuscript. All authors contributed to the discussion and revision of the manuscript.

## Competing interests

The authors declare no competing interests.
