## [Peer Review File · Nature Communications]

Reviewers' comments:

Reviewer #1 (Remarks to the Author):

Review

The manuscript entitled: "A Pressure-stabilized divalent ozonide CaO₃ and its impact on Earth's oxygen cycles" by Y. Wang et al. is devoted to the computational and experimental determination of a new form of calcium oxide. As I am not a theoretician, I will provide comments only on the experimental part of the manuscript.

General comment on geophysical implications:

The authors claim that the discovery of CaO₃ (with -2 oxidation state) is important for elucidating prominent seismic anomalies and oxygen cycles in Earth's interior but the proposed mechanisms for oxygen release and production of seismic anomalies are highly speculative. No quantitative information is provided to give a solid basis to their scenarios. This work is more a specific crystal chemistry study of calcium oxide polymorphs.

Comments on experimental results:

The authors claim the synthesis of a new Ca-O compound at high pressure and high temperature in a laser heated diamond anvil cell. This conclusion is solely based on in situ synchrotron X-ray diffraction. However, several points shed serious doubts about their claim:

- The authors have used off-line laser heating diamond anvil cell to synthesize new Ca-O compounds. This technique is very challenging, it is therefore fundamental that extreme care should be taken in the experimental approach as several aspects could lead to mistakes and wrong interpretation. However, very little technical details are provided to evaluate the correctness of the employed methodology in the Method Section. Exact sample purity (high purity is not a number)? Temperature measurements (~1800 K is not enough)? During chemical reactions the temperature strongly varies and can induce sample melting, reactions with the gasket material, etc. The authors should provide thermal emission spectra (in the full spectral range) to evaluate this critical point.

-The authors claim that no other compounds than CaO₃, CaO and O₂ are present in the sample chamber. It is not what can be deduced from the (only!) XRD pattern they provide. As shown in figure 1 (additional material), many extra-peaks that are not indexed are present in the XRD pattern (the extra-peaks are indicated by red arrows).

-More problematic: As shown in the figure 2 (additional material), the (200) and (111) reflections that are supposedly the main peaks of the new phase have totally different textures. The (200) is quasi-continuous while the (111) peak contains only few spots. This clearly indicates that these 2 peaks do not belong to the same phase. Also, if we look carefully at the (200) reflection, it has two and not only one component. The existence of a second component (indicated with a question mark in the figure below) explains the much larger peak width as compared to the (111) reflection.

-The peak intensities in figure 2c are very different from the ones in figure 2a. Where does this come from?

-This should be easy with the micro-beam of GSECARS but no in-situ mapping was performed (or provided) to determine the distribution of species in the sample chamber. No in situ Raman was carried out to characterize the newly formed chemical bonds. No chemical analysis of the recovered material was performed to demonstrate that no undesired chemical reactions happened.

All the elements mentioned here above shed serious doubts about the main claim of this work. I therefore do not recommend publication of this manuscript.

Figure 1.

Figure 2.

Reviewer #2 (Remarks to the Author):

The authors have used the CALYPSO method to predict the structure of CaO₃ at pressures that are relevant to the Earth's mantle. A CaO₃ phase, which contained O₃(²⁻) molecules, was predicted to be stable by ~30 GPa. The authors also considered finite temperature effects in determining stability, which is important in Earth conditions where the temperatures can be on the order of thousands of Kelvin, by using the quasiharmonic approximation. The phase contained closed shell bent O₃(²⁻) species, which is the first time that this novel oxidation state has been predicted. The computations inspired the synthesis of CaO₃ using high temperature high pressure synthesis techniques in a diamond anvil cell. The phase was characterized using XRD. The findings are discussed in the context of chemical reactions that could occur in the Earth's mantle.

This is a very nice manuscript that combines experiment and theory in the discovery of new solid state compounds. The text and analysis is very clearly written and presented. For this reason I suggest publication. I only have a few very minor suggestions.

1. Generally the writing is very well done, however one more proofread for the English should be carried out. For example it should be "... material structure and properties." The word "renovate" on the first page does not fit.
2. I do not understand the argument made regarding kinetics in the manuscript. To study kinetics it suggests that reaction paths may have been calculated to determine barriers. It might also imply that the phase is metastable at some pressure. However, I thought the phase was thermodynamically stable above 25.3 GPa (at 0 K), so why is the discussion of kinetics required?
3. Did the team look to see if the phase could be quenched to 1 atm (would it be dynamically stable at that pressure)? Did they try it experimentally?
4. Were the band structures calculated with HSE06 or with PBE? The Fermi level in Fig 3 is not clear. The authors could draw it in and compare the gap in the O₃(²⁻) species with the real CaO₃.
5. Do not use acronyms in the conclusions (HPHT).
6. The SI needs to be proofread. For example:
 - Ref 5 (Li) is mentioned twice
 - "Have been confirmed by (12-14)" is not a sentence.
 - The parentheses in the equation numbers are off.

Reviewer #3 (Remarks to the Author):

This manuscript describes results from a combined experimental and theoretical study of several calcium oxides at high pressures that results in a finding of a crystalline calcium ozonide (CaO₃). This calcium oxide finding is suggested to have implications for both reactions in the Earth's mantle and for seismic anomalies and oxygen cycles.

Other ozonides have been known (Chem. Ber. 129, 997 – 1001 (1996) and reference 17 of the manuscript for example. The authors demonstrate via structure prediction and optimization employing density functional theory followed with phonon calculations that CaO₃ should also be stable. They then prepared the CaO₃ in a diamond anvil cell that included laser heating to ~ 1800 K.

The calculations are well described and appear to be correct. The experiments are also well described and the analysis is clearly described. The authors also present possible reactions with minerals abundant in the mantle to support their suggestion of CaCO₃ existence.

I have therefore only several minor suggestions for the authors to improve the manuscript.

1) Authors should probably indicate and emphasize that the temperature and pressure in the mantle at the depths suggested is sufficient for the production of CaO₃ compared to the temperature and pressure used in the DAC experiment? The authors indicate that the reactions to produce CaCO₃ at about 20 GPa but is the temperature similar to the experimental temperature of 1800 K in the mantle boundary? Although this discussion is clear to this reviewer, it may need a remark in the manuscript. There may of course be other reactions not considered but the authors have considered abundant ones in the mantle.

2) Although the authors state on page 5 that pressures are considerably below the 50 GPa in the original search, the search was likely done at 0 K. Is this correct?

3) The lattice parameters are given on page 5 at presumably 35 GPa. This would agree with the values given for 30 GPa at 30 GPa in Table 1 of the supplementary Table 1. This could be emphasized for full clarity.

In summary, this manuscript contains a very interesting and plausible suggestion for the existence of a new calcium oxide compound (CaCO₃), calcium ozonide that may have implications for the chemistry and geophysics of the Earth's mantle. The originality of this suggestion and its examination with state-of-the-art theory and experiment support its publication in Nature Communications.

Response to the Reviewer #1

The manuscript entitled: "Pressure-stabilized divalent ozonide CaO₃ and its impact on Earth's oxygen cycles" by Y. Wang et al. is devoted to the computational and experimental determination of a new form of calcium oxide. As I am not a theoretician, I will provide comments only on the experimental part of the manuscript.

Authors' Reply: We thank the reviewer for a careful reading of our manuscript and the constructive comments and suggestions for its improvement.

General comment on geophysical implications:

The authors claim that the discovery of CaO₃ (with -2 oxydation state) is important for elucidating prominent seismic anomalies and oxygen cycles in Earth's interior but the proposed mechanisms for oxygen release and production of seismic anomalies are highly speculative. No quantitative information is provided to give a solid basis to their scenarios. This work is more a specific crystal chemistry study of calcibasis um oxide polymorphs.

Authors' Reply: In the present work, our extensive structure searches combined with energetic and dynamic stability evaluations predict the existence of crystalline CaO₃ and CaO₄ phases in at high pressure and high temperature conditions, leading to the discovery of these compounds by ensuing experimental synthesis and characterization. The newly discovered CaO₃ is the first and hitherto only known divalent ozonide, and its novel bonding character holds fundamental significance for probing exotic chemistry at high pressure and high temperature environments. The oxygen-rich CaO₃ also has great geoscience implications for assessing important processes inside Earth's deep interior, since CaO₃ has been synthesized at the temperature and pressure conditions relevant to those in Earth's key mantle regions. The reviewer is correct in stating that our main results are obtained in the context of studying calcium oxide polymorphs. We believe, however, that such results offer crucial new knowledge that has important implications for understanding prominent seismic anomalies and oxygen cycles inside Earth, thus providing a physics and/or chemistry basis to help elucidate prominent geoscience problems. It is within this interdisciplinary context that we are discussing the geoscience implications of our results, shedding light on the composition and properties of new constituents in Earth's interior.

Comments on experimental results:

The authors claim the synthesis of a new Ca-O compound at high pressure and high temperature in a laser heated diamond anvil cell. This conclusion is solely based on in situ synchrotron X-ray diffraction.

Authors' Reply: We thank the reviewer for raising this issue. To further confirm our results, we have repeated the experiment three times. Moreover, we also have included Raman spectra in our revised manuscript. The Raman peak near 767 cm^{-1} observed in experiment can be assigned to the stretching mode of O_3^{2-} and the high frequency mode of 1140 cm^{-1} is attributed to the stretching mode of intramolecular O–O in CaO_4 , whose bond length of 1.31 \AA is within the length range ($1.3\text{--}1.4\text{ \AA}$) of the superoxide O_2^- . These results reveal that CaO_3 and CaO_4 were simultaneously synthesized at high pressure and high temperature and contain the divalent ozonide group (O_3^{2-}) and superoxide group (O_2^-), respectively. Therefore, the synthesis of these new Ca-O compounds is confirmed by additional synchrotron XRD and in-situ Raman spectra.

However, several points shed serious doubts about their claim:

- The authors have used off-line laser heating diamond anvil cell to synthesize new Ca-O compounds. This technique is very challenging, it is therefore fundamental that extreme care should be taken in the experimental approach as several aspects could lead to mistakes and wrong interpretation. However, very little technical details are provided to evaluate the correctness of the employed methodology in the Method Section. Exact sample purity (high purity is not a number)? Temperature measurements ($\sim 1800\text{ K}$ is not enough)? During chemical reactions the temperature strongly varies and can induce sample melting, reactions with the gasket material, etc. The authors should provide thermal emission spectra (in the full spectral range) to evaluate this critical point.

Authors' Reply: We thank the reviewer for these constructive comments. In response, we have included more details on sample purity, temperature measurements and gasket materials etc. in the Methods section (see the content highlighted in blue color in revised manuscript).

Laser heating in DAC is widely used in material synthesis and characterization at high pressure and high temperature [Prakapenka, et al., High Pressure Res. 28, 225(2008); Goncharov et al., J. Synchrotron Rad. 16, 769 (2009); Dubrovinsky et al., J. Synchrotron Rad. 16, 737 (2009); Shen et al., PRL 92, 185701 (2004)]. In our present work, we utilized this

technique for synthesizing CaO_x compounds. This part of the experiment was performed by one of the coauthors, Liuxiang Yang, who has extensive working experience using laser heating at Max-Planck Institute at Mainz, Germany and Geophysical laboratory, Carnegie institute of Washington, as reflected in his recent works, see, e.g., *Rev. Sci. Instrum.* 83, 063905 (2012); *Nature* 534, 241 (2016); *Chin. Phys. B* 25, 076201 (2016); *Nat. Sci. Rev.* 4, 870 (2017); *Nat. Commun.* 8, 322 (2017); *Sci. Adv.* 6, 9405 (2020); *Sci. Adv.* 6, 9206 (2020).

In our reported experiment, we first tested the laser heating setup and calibrated temperature measurement using an Fe foil [see Fig. R1(a) and (b)], and the temperature was obtained by fitting the intensity versus wavelength dependence curve of emission from laser heated Fe to the well-known Plank radiation function [see, e.g., *Rev. Sci. Instrum.* 83, 063905 (2012)].

For the reported synthesis of CaO_x compounds, the heating was optically inspected to monitor temperature changes on the sample surface to identify the evidence that chemical reaction occurs while the laser power was gradually increased. Initially, at low laser powers, we did not see any temperature rising in the sample, which often happens during the heating of optically transparent samples. Continuing to increase the laser power and holding for 1 min, we saw the heating fluctuation and sudden heating runaway (an instant of very bright flash of light) in the cell. Such a sudden and rapid temperature rising is called a thermal runaway, which is a good sign of chemical reaction happening in our system. Similar thermal runaway has been seen in the study of melting of metallic Re and Pt [*Rev. Sci. Instrum.* 83, 063905 (2012); 84, 075111 (2013)]. For dielectric systems under pressure, the scenarios of thermal runaway also have been seen in laser-heated carbon nanotube within an oxygen environment [*J. Appl. Phys.* 107, 064319(2010)], the formation of high-pressure F-center defects in laser-heated KBr [*Phys. Rev. B* 97, 094103 (2018)], and chemical reactions in laser-heated alkaline-earth oxides and tungsten under pressure [*J. Phys. Chem. Solids* 70, 1117(2009)]. The thermal runaway phenomenon makes it very challenging to determine the sample temperature at the start of the chemical reaction under study. To estimate the heating temperature for laser-heated CaO_x , we performed a manual fitting to the Plank radiation function as soon as a bright flash of light was seen in the sample chamber indicating the occurrence of the chemical reaction in the DAC. The transient nature of this process makes it extremely challenging to pinpoint the reaction moment and temperature, as pointed out by the reviewer. Our fitting gives an estimated temperature of about 3,100 K [Fig. R1(c)] after the reaction indicated by a bright flash of light in the sample chamber was seen. The ensuing XRD and Raman measurements successfully identified the formation of CaO_3 and CaO_4

products, but the chemical reaction should have started at somewhat lower temperatures before the thermal runaway occurred, which is consistent with our calculated thermodynamic behaviors indicated in the P-T phase diagram.

Fig. R1. (a) A visual image of a laser heated Fe foil. (b) A typical measured spectrum for Fe in DAC heated for one minute fitted by the Planck radiation function. (c) A manual fitting to the Planck radiation function to determine the temperature right after the CaO and oxygen reaction has occurred.

The reviewer correctly points out that sometimes reactions inside DAC can cause temperature inhomogeneity, particularly when material melting is involved, and even lead to reaction with gasket materials [see, e.g., Nature Commun. 7, 13647 (2016)]. To examine possible sample reaction with gasket, we compared our collected XRD patterns and Raman spectra with available results on ReO_x ($x=1, 2, 3$) and O_2 after laser heating at high pressure, and found that these results are totally different from our observations. Moreover, we also switched to use an Fe gasket from the originally used Re foil, and we saw no difference between the measured results using these two different gasket materials [see the Raman spectra in Fig. 2a in the revised manuscript]. We reproduced the same products in three independent experimental runs and, on this basis, we are confident that the scenarios postulated by the reviewer did not occur in our experiment. Nevertheless, we thank the reviewer for raising this issue, which prompted us to carefully examine and rule out these other possibilities.

-The author claim that no other compounds than CaO_3 , CaO and O_2 are present in the sample chamber. It is not what can be deduced from the (only!) XRD pattern they provide. As shown in figure 1 (additional material), many extra-Peaks that are not indexed are present in the XRD pattern (the extra-peaks are indicated by red arrows).

Authors' Reply: We thank the reviewer for raising these important issues. It should be noted that beside the XRD patterns, we also have performed Raman spectroscopy measurements, as shown in Fig. 2a in the revised manuscript. We further carried out Rietveld refinements for the collected XRD patterns, and found that the XRD peaks can be well-fitted by contributions from CaO_3 , CaO_4 , CaO and oxygen.

Prompted by the reviewer's comments, and to ascertain our main finding of reported chemical reactions, we have reproduced our reaction products in three separate experimental runs using different reaction routes of CaO and oxygen, as well as Ca metal and oxygen with different (Re and Fe) gasket material. Furthermore, we also collected in-situ Raman mapping after heating to monitor the reaction products (see the collected Raman spectra shown below). The obtained Raman spectra has been added in Fig. 2a in the revised manuscript. Meanwhile, we also collected the XRD patterns (Fig. R2a) from several positions around the heating area at BL15U1, Shanghai Synchrotron Radiation Source. These newly collected XRD patterns are also shown in the figure below. Besides the starting CaO and epsilon oxygen, we also observed new reflections in the XRD patterns, reconfirming the reported chemical reactions, which is consistent with our previously shown results. Meanwhile, due to the presence of mixed phases, we performed Le Bail refinement using the predicted candidates for this sample shown in Fig. R2(b). One can see that the mixed phases of CaO_3 , CaO_4 , CaO and oxygen are present in the reaction products. These results affirm that the reported oxygen-rich phases have formed in our laser-heated high-pressure experiments.

Fig. R2. (a) The XRD patterns collected at different spots around the heating area in the cell.

(b) The Le Bail fitting with predicted phases is performed for the selected position. The lattice parameters obtained from the fitting are close to the data presented in the manuscript.

-More problematic: As shown in the figure 2 (additional material), the (200) and (111) reflections that are supposedly the main peaks of the new phase have totally different textures. The (200) is quasi-continuous while the (111) peak contains only few spots. This clearly indicates that these 2 peaks do not belong to the same phase. Also, if we look carefully at the (200) reflection, it has two and not only one component. The existence of a second component (indicated with a question mark in the figure below) explains the much larger peak width as compared to the (111) reflection. The peak intensities in figure 2c are very different from the ones in figure 2a. Where does this come from?

Authors' Reply: We thank the reviewer for raising these questions. In response, we carefully double-checked the obtained XRD patterns and did the Rietveld fitting, and the fitting results are shown in Fig. 2b in the revised manuscript. We find that the (200) and (111) reflections are still the main peaks of CaO₃. The quasi-continuous (200) peak also contains contribution from epsilon oxygen. Furthermore, the small peaks pointed out by the reviewer may have come from the metastable CaO₄. On the issue of the peak intensity, the results presented in the previous Fig. 2c and Fig 2a are similar, but the difference is that Fig. 2c shows the patterns collected during decompression. It is known that decompression may lead to deviations of x-ray beam from the heating position, therefore causing the mentioned issue. Again, to ascertain that our reported chemical reaction indeed occurred under the stated pressure-temperature conditions, we have reproduced our reaction products in three separate experimental runs using different reaction routes of CaO and oxygen, as well as Ca metal and oxygen with different (Re and Fe) gasket materials. In-situ Raman mapping after heating was also collected to monitor the reaction products, and the results are shown in Fig. 2a in the revised manuscript.

-This should be easy with the micro-beam of GSECARS but no in-situ mapping was performed (or provided) to determine the distribution of species in the sample chamber. No in situ Raman was carried out to characterize the newly formed chemical bonds. No chemical analysis of the recovered material was performed to demonstrate that no undesired chemical reactions happened. All the elements mentioned here above shed serious doubts about the main claim of this work. I therefore do not recommend publication of this manuscript.

Authors' Reply: We thank the reviewer for raising these pertinent issues, which indeed need exploration and clarification. In response, we have performed in-situ Raman mappings in different experimental runs involving CaO and oxygen, Ca metal and oxygen, and pure oxygen to examine the reaction products, and the obtained results are shown in Fig. 2 in the revised manuscript. Results from these experimental runs clearly show that the reaction products are reliably reproducible. In particular, the obtained products are well identified and their Raman spectra clearly distinct from the pure oxygen Raman spectrum. Since the oxygen-rich phases are unstable at ambient pressure, a chemical composition analysis at ambient conditions is ruled out. The Raman features are highly sensitive for the evaluation of the chemical species in the cell. In the revised manuscript, the newly obtained Raman mapping and XRD patterns are added to fingerprint the presence of the reported oxygen-rich phases in the Ca-O system.

Response to the Reviewer #2

Reviewer #2 (Remarks to the Author):

The authors have used the CALYPSO method to predict the structure of CaO₃ at pressures that are relevant to the Earth's mantle. A CaO₃ phase, which contained O₃(2-) molecules, was predicted to be stable by ~30 GPa. The authors also considered finite temperature effects in determining stability, which is important in Earth conditions where the temperatures can be on the order of thousands of Kelvin, by using the quasiharmonic approximation. The phase contained closed shell bent O₃(2-) species, which is the first time that this novel oxidation state has been predicted. The computations inspired the synthesis of CaO₃ using high temperature high pressure synthesis techniques in a diamond anvil cell. The phase was characterized using XRD. The findings are discussed in the context of chemical reactions that could occur in the Earth's mantle.

This is a very nice manuscript that combines experiment and theory in the discovery of new solid state compounds. The text and analysis is very clearly written and presented. For this reason I suggest publication. I only have a few very minor suggestions.

Authors' Reply: We thank the reviewer for a careful reading and positive assessment of our reported work.

1. Generally the writing is very well done, however one more proofread for the English should be carried out. For example it should be "... material structure and properties." The word "renovate" on the first page does not fit.

Authors' Reply: We have polished the English and modified the corresponding statements according to the reviewer's suggestions.

2. I do not understand the argument made regarding kinetics in the manuscript. To study kinetics it suggests that reaction paths may have been calculated to determine barriers. It might also imply that the phase is metastable at some pressure. However, I thought the phase was thermodynamically stable above 25.3 GPa (at 0 K), so why is the discussion of kinetics required?

Authors' Reply: We thank the reviewer for this comment. It is indeed true that the CaO_3 phase is thermodynamically stable above 25.3 GPa at 0 K. We have removed the statements regarding kinetics in the revised manuscript.

3. Did the team look to see if the phase could be quenched to 1 atm (would it be dynamically stable at that pressure)? Did they try it experimentally?

Authors' Reply: As shown in Supplementary Figure 7, the XRD signals for CaO_3 become weaker in intensity with decreasing pressure during decompression and disappear at 20 GPa, where the decompressed sample returns to the CaO phase. Thus, the CaO_3 phase cannot be quenched to ambient conditions.

4. Were the band structures calculated with HSE06 or with PBE? The Fermi level in Fig 3 is not clear. The authors could draw it in and compare the gap in the $\text{O}_3(2-)$ species with the real CaO_3 .

Authors' Reply: We thank the reviewer for these suggestions. The band structures of CaO_3 were calculated using the HSE functional and the details of the calculation were provided in Supplementary Information. We have updated Fig. 3 with a clearly indicated Fermi level. The bandgap of the isolated group of $\text{O}_3(2-)$ cannot be directly modelled by the energy band theory. So the MO-scheme for O_3^{2-} and a hypothetical periodic model of Ca_0O_3 were employed to calculate the electronic properties of $\text{O}_3(2-)$.

5. Do not use acronyms in the conclusions (HPHT).

Authors' Reply: We have made revision according to the reviewer's suggestion.

6. The SI needs to be proofread. For example:

- Ref 5 (Li) is mentioned twice
- "Have been confirmed by (12-14)" is not a sentence.
- The parentheses in the equation numbers are off.

Authors' Reply: We have made corrections following the reviewer's suggestions.

Response to the Reviewer #3

Reviewer #3 (Remarks to the Author):

This manuscript describes results from a combined experimental and theoretical study of several calcium oxides at high pressures that results in a finding of a crystalline calcium ozonide (CaO₃). This calcium oxide finding is suggested to have implications for both reactions in the Earth's mantle and for seismic anomalies and oxygen cycles. Other ozonides have been known (Chem. Ber. 129, 997 2013; 1001 (1996) and reference 17 of the manuscript for example. The authors demonstrate via structure prediction and optimization employing density functional theory followed with phonon calculations that CaO₃ should also be stable. They then prepared the CaCO₃ in a diamond anvil cell that included laser heating to ~ 1800 K. The calculations are well described and appear to be correct. The experiments are also well described and the analysis is clearly described. The authors also present possible reactions with minerals abundant in the mantle to support their suggestion of CaCO₃ existence.

I have therefore only several minor suggestions for the authors to improve the manuscript.

Authors' Reply: We thank the reviewer for a careful reading and positive assessment of our report work.

1) Authors should probably indicate and emphasize that the temperature and pressure in the mantle at the depths suggested is sufficient for the production of CaO₃ compared to the temperature and pressure used in the DAC experiment? The authors indicate that the reactions to produce CaO₃ at about 20 GPa but is the temperature similar to the experimental temperature of 1800 K in the mantle boundary? Although this discussion is clear to this reviewer, it may need a remark in the manuscript. There may of course be other reactions not considered but the authors have considered abundant ones in the mantle.

Authors' Reply: We thank the reviewer for raising these issues and the constructive comments. We have performed new measurements and analysis of the laser-heating assisted synthesis of CaO_x, and amended the corresponding presentation and discussion in the revised manuscript.

2) *Although the authors state on page 5 that pressures are considerably below the 50 GPa in the original search, the search was likely done at 0 K. Is this correct?*

Authors' Reply: It is correct that we followed the standard practice of performing the initial structure search at 0 K. The thermal effects are accounted for in the ensuing calculations via further examining vibrational contributions and entropic effects for the relevant phases, which resulted in the construction of the finite-temperature phase diagram of CaO₃ in Fig. 1(b).

3) *The lattice parameters are given on page 5 at presumably 35 GPa. This would agree with the values given for 30 GPa at 30 GPa in Table 1 of the supplementary Table 1. This could be emphasized for full clarity.*

Authors' Reply: We have revised data on lattice parameters in Supplementary Table 1 following the reviewer's suggestion.

In summary, this manuscript contains a very interesting and plausible suggestion for the existence of a new calcium oxide compound (CaCO₃), calcium ozonide that may have implications for the chemistry and geophysics of the Earth's mantle. The originality of this suggestion and its examination with state-of-the-art theory and experiment support its publication in Nature Communications.

Authors' Reply: We appreciate the reviewer's positive assessment of our work and support for the publication of the present manuscript in Nature Communications.

REVIEWERS' COMMENTS:

Reviewer #1 (Remarks to the Author):

The authors have made a significant effort to answer the experimental issues I raised in my previous review. As requested, they provide additional experimental details, XRD and Raman data that support their conclusions. I consider that their manuscript is largely improved and deserves publication in Nature Communications.

Reviewer #2 (Remarks to the Author):

The authors have revised their manuscript so that they have answered all of my questions adequately. Since I am not an experimentalist, I cannot determine if the authors were able to satisfy the critical referee. However, their response letter appears to have fully taken the referee's comments into consideration.

Reviewer #3 (Remarks to the Author):

This revised manuscript reports on a combined experimental and theoretical study of a possible ozonide compound CaO_3 formed at high pressure and temperature from CaO and oxygen. The authors have carried out additional x-ray diffraction experiments and have now added Raman spectroscopy to their methods. The experiments were carried out employing diamond anvil cell methods with synchrotron x-ray diffraction and Raman spectroscopy together with standard methods to monitor the sample pressure and temperature. The experiments were carried out by expert high-pressure experimentalists. The theory component of the manuscript was prepared by researchers with past experience with density functional theory (DFT) applied to condensed matter problems similar to the one in this study. The main question is whether the interpretation of the results can be accepted as suggested by the authors. Although ozonide compounds are well known, this is the first report of CaO_3 and it is well supported by the experiments and DFT results presented.

This reviewer feels that, with the addition of Raman spectroscopy and repeating the experiments, the experimental details have been improved to that of a reasonably expected expert level. Although the direct connection to seismic anomalies is somewhat speculative, the ideas presented regarding this together with the experimental and theory results can provide a very good starting point for future work.

CaO_3 identity is supported by Raman active mode frequencies that agree with DFT calculations. The Raman frequencies agreement with DFT calculations is indicated in figure 2 of the revised manuscript. The synchrotron x-ray diffraction data is shown to agree with the predicted structure of CaO_3 and this agreement is well illustrated in figure 2 of the revised manuscript.

Although a possible weak point of the manuscript is with the DFT calculations and structure predictions that depend on correct choices of pseudopotential functionals. The consistency achieved with experiment however supports the authors claim for CaO_3 if this well-known limitation of DFT is kept in mind.

There are just a few minor points for the authors that would add clarity to the manuscript.

- 1) The pressure for SM Fig. 8 b should be clearly indicated in the caption or on the figure.
- 2) It may be useful to add just a few more details such as an emphasis on the unit cell size required for the diffraction pattern predicted. Is the 2 fu structure suggested the only structure consistent with the experimental findings?
- 3) It may be useful to put a list of predicted diffraction pattern details for the CaO_3 in the Supplementary Information although these are easily obtained from the structural details given in

Table 1. This would just provide immediate information for readers.

In summary, this reviewer feels that the originality of the manuscript that employed a well-known structure search method, together with both the detailed and careful experimental details presented together with the DFT calculations described supports publication in this journal.

Reply to the Reviewer Comments

REVIEWERS' COMMENTS:

Reviewer #1 (Remarks to the Author):

The authors have made a significant effort to answer the experimental issues I raised in my previous review. As requested, they provide additional experimental details, XRD and Raman data that support their conclusions. I consider that their manuscript is largely improved and deserves publication in Nature Communications.

Authors' Reply: We thank the reviewer for the careful reading and positive assessment of our report work.

Reviewer #2 (Remarks to the Author):

The authors have revised their manuscript so that they have answered all of my questions adequately. Since I am not an experimentalist, I cannot determine if the authors were able to satisfy the critical referee. However, their response letter appears to have fully taken the referee's comments into consideration.

Authors' Reply: We thank the reviewer for the positive assessment of our reported work.

Reviewer #3 (Remarks to the Author):

This revised manuscript reports on a combined experimental and theoretical study of a possible ozonide compound CaO_3 formed at high pressure and temperature from CaO and oxygen.

The authors have carried out additional x-ray diffraction experiments and have now added Raman spectroscopy to their methods. The experiments were carried out employing diamond anvil cell methods with synchrotron x-ray diffraction and Raman spectroscopy together with standard methods to monitor the sample pressure and temperature. The experiments were carried out by expert high-pressure experimentalists. The theory component of the manuscript was prepared by researchers with past experience with density functional theory (DFT) applied to condensed matter problems similar to the one in this study. The main question is whether the interpretation of the results can be accepted as suggested by the

authors. Although ozonide compounds are well known, this is the first report of CaO_3 and it is well supported by the experiments and DFT results presented.

This reviewer feels that, with the addition of Raman spectroscopy and repeating the experiments, the experimental details have been improved to that of a reasonably expected expert level. Although the direct connection to seismic anomalies is somewhat speculative, the ideas presented regarding this together with the experimental and theory results can provide a very good starting point for future work.

CaO_3 identity is supported by Raman active mode frequencies that agree with DFT calculations. The Raman frequencies agreement with DFT calculations is indicated in figure 2 of the revised manuscript. The synchrotron x-ray diffraction data is shown to agree with the predicted structure of CaO_3 and this agreement is well illustrated in figure 2 of the revised manuscript. Although a possible weak point of the manuscript is with the DFT calculations and structure predictions that depend on correct choices of pseudopotential functionals. The consistency achieved with experiment however supports the authors claim for CaO_3 if this well-known limitation of DFT is kept in mind.

Authors' Reply: We thank the reviewer for the careful reading and positive assessment of our reported work.

There are just a few minor points for the authors that would add clarity to the manuscript.

1) The pressure for SM Fig. 8 b should be clearly indicated in the caption or on the figure.

Authors' Reply: We thank the reviewer for a careful reading of our manuscript. We have added the pressure of 37.4 GPa in the caption of Supplementary Figure 8b.

2) It may be useful to add just a few more details such as an emphasis on the unit cell size required for the diffraction pattern predicted. Is the 2 fu structure suggested the only structure consistent with the experimental findings?

Authors' Reply: We thank the reviewer for the constructive suggestions. We performed structural searches at 30 and 50 GPa for CaO_3 with one to four formula units and more than two thousand candidate structures have been assessed based on total energy calculations. We find that the most stable structure of CaO_3 predicted by our structure searches adopts $P-42_1m$ symmetry, which contains 2 f.u. per cell. It is known that thermodynamics plays a critical

role in determining the structures in practical environments. In particular, the synchrotron x-ray diffraction data and Raman spectroscopy agree with the predicted structure of CaO_3 . On this basis, we assign the 2 f.u. structure for the experimentally synthesized CaO_3 .

3) It may be useful to put a list of predicted diffraction pattern details for the CaO_3 in the Supplementary Information although these are easily obtained from the structural details given in Table 1. This would just provide immediate information for readers.

Authors' Reply: We thank the reviewer for these constructive comments. We have added the diffraction pattern of the predicted structure in $P-42_1m$ symmetry in Supplementary Figure 7b.

In summary, this reviewer feels that the originality of the manuscript that employed a well-known structure search method, together with both the detailed and careful experimental details presented together with the DFT calculations described supports publication in this journal.

Authors' Reply: We appreciate the reviewer's positive assessment of our work and support for the publication of the present manuscript in Nature Communications.